# A Novel Selective 11β-HSD1 Inhibitor, (E)-4-(2-(6-(2,6-Dichloro-4-(Trifluoromethyl)Phenyl)-4-Methyl-1,1-Dioxido-1,2,6-Thiadiazinan-2-yl)Acetamido)Adamantan-1-Carboxamide (KR-67607), Prevents BAC-Induced Dry Eye Syndrome

**DOI:** 10.3390/ijms21103729

**Published:** 2020-05-25

**Authors:** Yoon-Ju Na, Kyoung Jin Choi, Won Hoon Jung, Sung Bum Park, Sein Kang, Jin Hee Ahn, Ki Young Kim

**Affiliations:** 1Therapeutics & Biotechnology Division, Korea Research Institute of Chemical Technology, 141 Gajeong-ro, Yuseong-gu, Daejeon 34114, Korea; yjna@krict.re.kr (Y.-J.N.); choikj@krict.re.kr (K.J.C.); whjeung@krict.re.kr (W.H.J.); spark@krict.re.kr (S.B.P.); sein@krict.re.kr (S.K.); 2Department of New Drug Discovery and Development, Chungnam National University, 99 Daehak-ro, Yuseong-gu, Daejeon 34134, Korea; 3Department of Chemistry, Gwangju Institute of Science and Technology, Gwangju 61005, Korea; jhahn@gist.ac.kr

**Keywords:** dry eye syndrome, 11β-hydroxysteroid dehydrogenase 1 (11β-HSD1), reactive oxygen species (ROS)

## Abstract

Dry eye syndrome is the most common eye disease and it is caused by various reasons. As the balance of the tear film that protects the eyes is broken due to various causes, it becomes impossible to properly protect the eyes. In this study, the protective effects and underlying mechanisms of topical (E)-4-(2-(6-(2,6-dichloro-4-(trifluoromethyl)phenyl)-4-methyl-1,1-dioxido-1,2,6-thiadiazinan-2-yl)acetamido)adamantan-1-carboxamide (KR-67607), a novel selective 11β-hydroxysteroid dehydrogenase 1 (11β-HSD1) inhibitor, were investigated in benzalkonium chloride (BAC)-induced dry eye syndrome. BAC-treated rat eyes induced significant increases in ocular surface damage, decreased corneal thickness, corneal basement membrane destruction in the conjunctival epithelium, and expression of pro-inflammatory cytokines tumor necrosis factor-α and 11β-HSD1. These effects of BAC were reversed by topical KR-67607 treatment. Furthermore, KR-67607 decreased 4-hydroxynonenal expression and increased antioxidant and mucus secretion in BAC-treated rat eyes. Taken together, a novel selective 11β-HSD1 inhibitor can prevent BAC-induced dry eye syndrome by inhibiting pro-inflammatory cytokine and reactive oxygen species expression via the inhibition of both 11β-HSD1 activity and expression.

## 1. Introduction

Dry eye syndrome (DES) is the most common ophthalmologic disease [1]. There are various causes of DES. Either the tear film component that protects the surface of the eye is excessively evaporated or there is a lack of secretion in the eyes, causing discomfort, redness, and burning sensation, thus resulting in DES [2]. Continued progression of DES can cause inflammation, which can cause problems with cornea or conjunctiva and finally could have a bad effect on the vision [3].

Glucocorticoids are steroid hormones that mediate immunological and inflammatory responses, energy metabolism, cardiovascular homeostasis, and general responses to stress [4,5,6]. Their intracellular levels are mainly regulated by 11β-hydroxysteroid dehydrogenase (11β-HSD) enzymes [7,8]. 11β-HSD1 is an enzyme that converts inactive cortisone into active cortisol [8]. Overexpression of the glucocorticoid enzyme cortisol has been associated with type 2 diabetes, obesity, osteoporosis, arthritis, myopathy, and inflammatory diseases [9,10,11]. Recently, 11β-HSD1, which is expressed in human and rodent eyes, has been studied as a target of glaucoma in human and rat ocular tissues. The concentration of 11β-HSD1 was increased in both glaucoma animal models and glaucoma patients. The 11β-HSD1 inhibitor was effective in reducing the intraocular pressure, which is the main objective of glaucoma treatment [12,13,14].

Reactive oxygen species (ROS) is a by-product of oxygen metabolism [15]. There are many kinds of ROS, such as superoxide, hydrogen peroxide, and hydroxyl radical, which can damage tissues in vivo [15,16]. This oxidative stress occurs due to the imbalance between anti-oxidant and pro-oxidant substances [15]. Oxidative stress is reportedly associated with various inflammatory diseases, such as cataract, macular degeneration, uveitis, keratitis, and DES [17,18]. In the eyes, oxidative damage caused by excessive ROS leads to the inflammation of conjunctiva and cornea, which prevents the formation of a tear film that protects the eyes. Thus, it is important to control the amount of ROS in the eye [19]. Furthermore, several studies have investigated the relationship between 11β-HSD1 and ROS [20]. Many reports show that the expression of 11β-HSD1 and ROS is involved in various disease models [12,20,21]. 

Previously, we showed the pathogenic relationships between 11β-HSD1 and benzalkonium chloride (BAC)-induced DES in rat eyes and human conjunctival epithelial cells and proposed the 11β-HSD1 as a novel therapeutic target for prevention and/or treatment of DES, also carbenoxolone (CBX), a non-selective 11β-HSD1 inhibitor, showed therapeutic effects [22]. In this study, we aimed to investigate the protective effects of a novel selective 11β-HSD1 inhibitor, (E)-4-(2-(6-(2,6-dichloro-4-(trifluoromethyl)phenyl)-4-methyl-1,1-dioxido-1,2,6-thiadiazinan-2-yl)acetamido)adamantan-1-carboxamide (KR-67607) (Figure 1) [23] and the role of 11β-HSD1 in eyes with BAC-induced injury. The results showed the protective effect of the 11β-HSD1 inhibitor against oxidative stress-mediated cellular damages and the role of 11β-HSD1 on anti-oxidant pathway during BAC-induced eye injury.

## 2. Results

### 2.1. Effects of 11β-HSD1 Inhibitors on Ocular Surface Damage in BAC-Treated SD Rat Eyes

Animals underwent the following eight treatments: control, 0.2% BAC, 0.2% BAC + 0.75 mg/mL KR-67607, only 0.75 mg/mL KR-67607, 0.2% BAC + 1.5 mg/mL KR-67607, only 1.5 mg/mL KR-67607, 0.2% BAC + 1 mg/mL CBX, or only 1 mg/mL CBX treatment. Compounds were topically administered twice daily at 09:30 and 17:30 for 10 days.

In previous studies, we identified ocular surface damage by BAC using rose bengal staining [22]. Herein, the eyes were pretreated with drugs for three days and with drug and BAC for seven days. At the last day, ocular surface damage by BAC was identified by rose bengal staining. BAC has been used as a preservative in ophthalmic solutions [24]. However, long-term use of BAC is known to cause inflammation and DES [25,26]. Thus, we used BAC as an inducer for DES. 

When BAC is administered, damage is inflicted on the ocular surface, and the rose bengal solution adheres to the wound surface of the eyeball and becomes reddish. In this study, CBX, a known inhibitor of 11β-HSD1, was used as a reference compound. Unlike the group treated with BAC alone, the group treated with BAC plus KR-67607 showed surface damage recovery in a concentration-dependent manner (Figure 2a). In the group treated with only the drug alone (KR-67607 or CBX), the tendency of without damages, similar to that in the control group. KR-67607 and CBX were found to be effective in improving the wound healing on the ocular surface as confirmed by scoring on rose bengal staining (Figure 2b). In this experiment, we confirmed that KR-67607 had a protective effect for the cornea in the BAC-induced DES model.

### 2.2. Effects of 11β-HSD1 Inhibitors on Corneal Epithelial and Basement Membrane Thickness in BAC-Treated SD Rat Eyes

Corneal epithelial thickness tends to decrease in patients with DES. The tear film does not adequately protect the corneal surface in such patients as its thickness decreases due to the injury to the corneal epithelium [27]. In this experiment, the thickness of the corneal epithelium was found to be decreased in the BAC-treated group. Moreover, the corneal thickness was significantly increased in the BAC-treated group that was treated with KR-67607 and CBX simultaneously. The thickness of the epithelial cells was almost similar in the group treated only with the drug (Figure 3a), which is expressed statistically in Figure 3b.

The basement membrane was stained with periodic acid-Schiff’s (PAS) staining. The basement membrane functions to supply the whole cornea with nutrients while keeping other cells in good condition. Thus, when the basement membrane is injured, the supply of nutrients to the eyes is compromised and the condition worsens. In the group treated with BAC, the basement membrane shape was not formed well, and the dyeing looks pale. In the group treated with BAC plus KR-67607, BAC-induced basement membrane damage was restored, and the staining was clearer. In the group treated with only the drug, it tended to be similar to or slightly more reddish than the control (Figure 3a). In this experiment, 11β-HSD1 inhibitors showed increased corneal thickness and basement membrane protective effects. 

### 2.3. Effects of 11β-HSD1 Inhibitors on Conjunctival ROS Level in BAC-Treated SD Rat Eyes

The expression level of 4-hydroxynonenal (4-HNE), a byproduct of ROS, was analyzed in the conjunctiva. 4-HNE is a type of lipid peroxide that is highly reactive and causes DNA fragmentation and protein modification [28,29,30]. Particularly in ophthalmic diseases, it is known to be associated with dry eye and conjunctivochalasis [30]. The expression of 4-HNE was increased in the BAC-treated group, which was significantly suppressed by KR-67607 (0.75 mg/mL and 1.5 mg/mL) (Figure 4a). Furthermore, the expression of superoxide dismutase 1 (SOD1), an antioxidant enzyme, was also measured (Figure 4b). In contrast to 4-HNE, the expression of SOD1 was decreased in the BAC group, which was increased by KR-67607 or CBX. These results show that 11β-HSD1 inhibitors suppress BAC-induced oxidative stress by increasing the expression of antioxidant enzyme.

### 2.4. Effects of 11β-HSD1 Inhibitors on Conjunctival Pro-Inflammatory Marker Expression in BAC-Treated SD Rat Eyes

Inflammatory cytokines such as interleukin-6 (IL-6) and TNF-α are secreted by BAC. And it doesn’t wash out well, so it stays on my eyes for a long time. As a result, BAC causes inflammation, allergies, fibrosis and DES [22]. The expression level of TNF-α was checked to see if it inhibited inflammation in the conjunctiva by KR-67607. In the IHC results, 0.2% BAC induced TNF-α expression in the conjunctival epithelial cells, which was suppressed by 0.75 mg/mL and 1.5 mg/mL KR-67607 (Figure 5). These results suggest that the protection of KR-67607 against BAC-induced DES is associated with the protection of the corneal epithelium via the inhibition of TNF-α production. These results suggest that the protection of KR-67607 against BAC-induced DES is associated with the protection of the conjunctival epithelium via the inhibition of TNF-α production.

### 2.5. Effects of 11β-HSD1 Inhibitors on Mucus Secretion in Conjunctival Epithelium of BAC-Treated SD Rat Eyes

Conjunctiva’s goblet cells secrete mucus to protect the eyes, and mucus has the function of supporting immunity by clearing allergens and debris off the surface. In addition, if goblet cells do not secrete mucus properly, the tear film may not form, thus causing DES and ocular surface inflammatory diseases [31,32,33]. To examine whether the protective effects of KR-67607 against DES are associated with mucus secretion in the conjunctival epithelium, the effects of KR-67607 on mucus secretion were measured by PAS staining in the conjunctiva of SD rat eyes. BAC reduces mucus secretion in goblet cells, which is improved by KR-67607 (0.75 mg/mL and 1.5 mg/mL). (Figure 6). These results show that the protection of KR-67607 against BAC-induced DES is associated with the induction of mucus secretion.

### 2.6. Effects of 11β-HSD1 Inhibitors on 11β-HSD1 Expression in BAC-Treated SD Rat Eyes

Some studies have shown that the expression of 11β-HSD1 in the eye is associated with ophthalmological disease development [14,34,35]. Our previous study showed that CBX suppress BAC-induced 11β-HSD1 expression and activity [22]. In this study, 11β-HSD1 expression in conjunctival epithelial cells was confirmed by IHC (Figure 7). In the BAC-treated group, the expression of 11β-HSD1 increased. KR-67607 treatment decreased the 11β-HSD1 expression in conjunctival epithelial cells in BAC-treated SD rat eyes. It was assumed that increased 11β-HSD1 expression by BAC exacerbates DES by mediating an inflammatory response and reducing mucus secretion, which is improved by KR-67607.

## 3. Discussion

In our previous study, we synthesized a novel selective 11β-HSD1 inhibitor, (E)-4-(2-(6-(2,6-dichloro-4-(trifluoromethyl)phenyl)-4-methyl-1,1-dioxido-1,2,6-thiadiazinan-2-yl)acetamido) damantan-1-carboxamide (KR-67607) [23]. KR-67607 demonstrated protective effects in in vivo and in vitro glaucomatous models. KR-67607 ameliorated ischemia-reperfusion-induced eye damage in human trabecular meshwork cells and in an animal model. In addition, KR-67607 decreased intraocular pressure in episcleral vein ligation model. In addition, to investigate the selectivity and inhibitory effect of KR-67607, reductase activity of 11β-HSD1 was measured by overexpression of human and mouse 11β-HSD1 in CHO-K1 cells. IC_50_ value of human and mouse showed a low concentration (4 nM and 7 nM, respectively), which is lower than CBX. Furthermore, KR-67607 did not restrict enzymatic activity of 11βHSD2 and 17β-HSD1/2 (IC50 > 50 μM) [23]. In the last study, CBX, a non- selective inhibitor of 11β-HSD1, had a therapeutic effect on DES. CBX showed protective effect of the ocular surface, inflammation in cornea and cell death in conjunctival epithelium in DES animal models. In addition, mucus secretion was increased in conjunctival goblet cells to stabilize the tear film, thus alleviating DES [22]. Thus, we investigated the therapeutic effects of a novel selective 11β-HSD1 inhibitor, KR-67607, in BAC-induced DES model.

ROS are a byproduct of oxygen metabolism and include superoxide, hydrogen peroxide, hydroxyl radicals, and lipid peroxide, which are highly reactive and can damage tissues in vivo [36,37,38]. This oxidative stress occurs due to the imbalance between anti-oxidant and pro-oxidant substances. ROS stress has been studied in ophthalmic diseases [36]. Ocular oxidative stress is caused by ultraviolet rays or various external environments, and when it fails to control oxidative stress, it can cause various inflammatory diseases, such as cataract, macular degeneration, uveitis, and keratitis [19]. Therefore, controlling the oxidative stress in the eye is very important. Tears and conjunctiva have various antioxidant substances that protect the ocular surface from oxidative stress. However, when oxidative homeostasis is broken due to various causes, excessive ROS is generated, which adversely affects the tear film protecting the eye surface [39,40,41]. Several studies have investigated the relationship between 11β-HSD1 and ROS and have has shown that 11β-HSD1 is associated with the expression of ROS [9,20]. However, the association between 11β-HSD1 and ROS-induced DES has not been studied. In this study, we assumed that 11β-HSD1 produced ROS, which could cause DES. The effects of 11β-HSD1 and ROS on DES were investigated, and KR-67607, a novel selective 11β-HSD1, was found to improve DES.

DES was induced in a rat model by BAC treatment on the SD rat eyes. BAC has been used as a preservative in ophthalmic solutions. However, it is known that excessive or prolonged use can cause inflammation and DES in the eyes. Thus, many studies used BAC as a DES inducer [40,41]. The animal models were pretreated with the drug for three days and were then treated with the BAC and the drug or only the drug for seven days. The rose bengal staining method was used to confirm the damage to the ocular surface. Rose bengal dye is deposited on the wounded part of the ocular surface, which then appears in red [42]. In the group treated with BAC, various parts of the eyes were stained. This shows that BAC inflicted damage on the ocular surface. In the KR-67607-treated group, it was confirmed that the wound on the surface of the eye had begun healing, and in the group treated with only the drug, the surface had healed to appear like a normal ocular surface. Patients with DES tend to have decreased corneal epithelial thickness [27]. In the group treated with BAC, BAC reduced the thickness of the corneal epithelium. However, when KR-67607 was administered to animals with DES, the thickness of corneal epithelial cells normalized. We also stained the basement membrane of the cornea with PAS staining [43]. The basement membrane supports corneal epithelial cells and has a role in delivering nutrients. However, when the eyes have inflammation or DES, its shape gets distorted [44]. In this experiment, the thickness of the basement membrane with DES was thin or the membrane appeared deformed. However, in the KR-67607-treated groups, the thickness of the basement membrane increased. These results suggest that the KR-67607, a novel selective 11β-HSD1 inhibitor, could protect the corneal epithelium and help the basement membrane function properly.

The ROS produced in the eyes can cause DNA damage and produce lipid peroxides [19,45]. Various antioxidant enzymes, SOD1 and glutathione peroxidase 4, remove ROS, such as lipid peroxides, hydrogen peroxide, and superoxide [18]. However, it is known that BAC induced 4-HNE, a by-product of lipid peroxide, in conjunctival epithelial cells [46]. This indicates that oxidative stress can also be induced by BAC [47]. The expression of SOD1, an antioxidant enzyme, and 4-HNE, a lipid peroxide, was measured in conjunctival epithelial cells. The expression of SOD1 was decreased by BAC, but that of SOD1 was increased in epithelial cells in KR-67607-treated groups. Conversely, the expression of 4-HNE was increased by BAC, which was reversed by KR-67607. These results suggest that KR-67607 increases the expression of SOD1 and removes reactive ROS, such as lipid peroxide.

In addition, KR-67607 also showed an inflammation-alleviating effect on the conjunctiva. In the BAC-treated group, the expression of TNF-α was increased in conjunctival epithelial cells. Studies have shown that 11β-HSD1 mediates inflammation [9,10,11]. Moreover, pro-inflammatory cytokines, such as IL-1, IL6, and TNF-α, are elevated in DES. In addition, these pro-inflammatory cytokines contribute to DES and cause corneal epithelial cell death [48,49]. Our previous studies also showed a decrease in 11β-HSD1 and inflammatory cytokines by CBX in the BAC-treated group [22]. The present study showed BAC-induced pro-inflammatory cytokine expression in conjunctival epithelium, which was suppressed by KR-67607 treatment. KR-67607 also showed pro-inflammatory cytokine inhibition in the BAC-induced DES model.

Ocular surface goblet cells are scattered along the conjunctival epithelium. Conjunctiva has goblet cells that produce mucus in the mucous layer, a component of the tear film. The mucus secreted by goblet cells has anti-adhesive, lubricating, water retention, and pathogen barrier functions [50]. If the mucus is not properly secreted, the tear film collapses, thus resulting in DES. In the group treated with BAC, mucin secretion was inhibited, which was improved by treatment with KR-67607. These result suggests that KR-67607 protects the conjunctival epithelium which assists goblet cells to secrete mucus properly. 

Lastly, BAC increased the expression of 11β-HSD1 as noted in IHC results; this was reversed by KR-67607 treatment. The reduced expression of 11β-HSD1 by KR-67607 is expected to be associated with ocular surface protection, increased corneal epithelial and basal cell thickness, decreased inflammation and ROS, and increased mucus secretion. However, the exact mechanism for this needs further research.

## 4. Materials and Methods

### 4.1. Animals and Drug Administration

All animal experiments were conducted using Sprague-Dawley (SD) rats (Orient Bio Inc., Seongnam, Korea). We have been approved by the *Guidelines for Animal Experimentation under admission of the Institutional Animal Care and Use Committee* of the Korea Research Institute of Chemical Technology (Approval code; 2018-7A-02-02). Animals were housed in the specific pathogen-free conditions with 12-h:12-h light/dark cycle, temperature of 23 ± 1 °C, ventilated 10–12 times per hour, and with a humidity of 55% ± 5%. Rats were provided food and water *ad libitum*.

0.2% BAC (Sigma-Aldrich, St. Louis, MO, USA) was used to induce DES. We dissolved the KR-67607 or CBX (Amifinecom Inc., Pertersburg, VA, USA) in 5% dimethyl sulfoxide (DMSO; Sigma-Aldrich, St. Louis, MO, USA). Animals underwent the following eight treatments: Control, 0.2% BAC, 0.2% BAC + 0.75-mg/mL KR-67607, only 0.75-mg/mL KR-67607, 0.2% BAC + 1.5-mg/mL KR-67607, only 1.5-mg/mL KR-67607, 0.2% BAC + 1-mg/mL CBX, or only 1-mg/mL CBX treatment. Drugs were administrated topically (one drop, 20 μL) twice a day (09:30, 17:30) for 10 days. 

### 4.2. Rose Bengal Staining and Scoring

The SD rats were treated with 1ul of 1% rose Bengal dye solution (Sigma-Aldrich, St. Louis, MO, USA) on the eyes. Artificially, the eyes were blinked to make the dye spread well. After 1 min, the ocular surface damage was observed under a microscope (Nikon, Tokyo, Japan). We used the scoring method suggested by Takamura et al. (2012) [51]: three sections of the ocular surface nasal bulbar conjunctiva (1+ area), cornea (2+ area), and temporal bulbar conjunctiva (3+ area). Each compartment was graded as 0 (no damage), 1 (partial damage), 2 (damage in more than half the area), or 3 (damage in the entire area), with a maximum possible score of nine points. 

### 4.3. Histology

The cornea and conjunctiva were fixed in the Davison’s fixative solution (Sigma-Aldrich, St. Louis, MO, USA) for 24 h. The samples were washed in tap water for 4 h, and paraffin embedding was performed with a tissue processing machine (Leica TP 1020, Wetzlar, Germany). After that, they were cut to 5-μm using a microtome (Leica, Wetzlar, Germany), and then stainings were performed. All slide figures were taken with a DS-Ri2 microscope (Nikon, Tokyo, Japan).

### 4.4. Immunohistochemistry (IHC) Staining

IHC staining procedures followed the protocol provided by Abcam’s IHC-Paraffin Protocol (Cambridge, MA, USA). After fixation, samples were deparaffinized and hydrated. Then, antigen retrieval was performed with Tris-EDTA buffer (10-mM Tris base and 1-mM EDTA solution, pH 8.0) + 0.05% Tween20. Sections were incubated in 10% normal goat serum (Santa Cruz, Dallas, TX, USA) in 1% bovine serum albumin to block non-specific binding; this was followed by treatment with TNF-α (1:500 dilution) (Abcam, Cambridge, MA, USA), 4-HNE (1:500 dilution) (Abcam, Cambridge, MA, USA), SOD 1 (Abcam, Cambridge, MA, USA), and 11β-HSD1 (1:500 dilution) (Santa Cruz, Dallas, TX, USA) antibody overnight at 4 °C. 24 h later. After washing slides with Tris-buffered saline with 0.025% Tritons X-100, 0.3% hydrogen peroxide was administered for 15 min. Subsequently, secondary antibody (Anti-rabbit IgG, Santa Cruz, Dallas, TX, USA) was administered for 1h. Next, chromogen (Sigma-Aldrich, St. Louis, MO, USA) was administered for 10-15 min. Then the slide was washed in running water, and then counterstained with hematoxylin.

### 4.5. Periodic Acid and Schiff’s (PAS) Staining

The PAS staining protocol followed the method provided by Novaultra™ (IHC World, Woodstock, MD, USA). Briefly, samples were deparaffinized, hydrated, and then oxidized by adding 0.5% periodic acid for 5 min. After oxidation, samples were washed in distilled water for 3 min and put in Schiff reagent for 15 min. Then, the slides were rinsed in warm tap water for 5 min. Lastly, Mayer’s hematoxylin was used for counterstaining.

### 4.6. Statistics

Statistical analyses were conducted using GraphPad Prism software (GraphPad Software Inc., La Jolla, CA, USA). The results are expressed as the means ± S.E.M. Statistical significance was determined by Student’s *t* test or a one-way analysis of variance (ANOVA) followed by Tukey’s multiple-comparison test. *P* < 0.05 was considered statistically significant.

## 5. Conclusions

Topical KR-67607 treatment suppresses DES progression by preventing ocular surface damage and protecting the corneal epithelium and basement membrane by inhibiting pro-inflammatory cytokine and ROS expression. A novel selective 11β-HSD1 inhibitor, KR-67607, shows potential therapeutic approach to the treatment and/or prevention of DES.

## Figures and Tables

**Figure 1 ijms-21-03729-f001:**
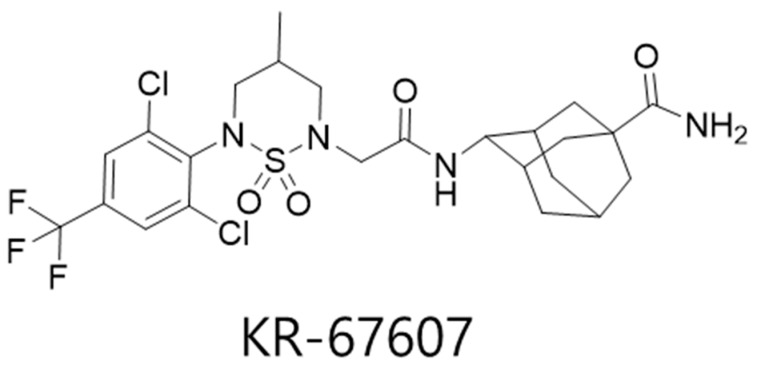
Chemical structure of KR-67607.

**Figure 2 ijms-21-03729-f002:**
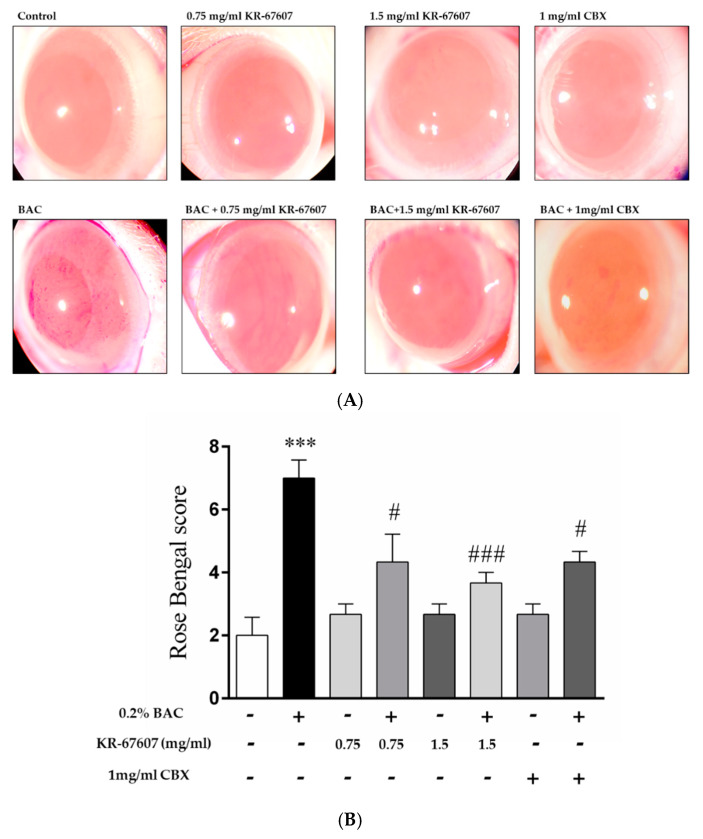
Effects of 11β-HSD1 inhibitors on ocular surface damage in BAC-treated SD rat eyes. Animals underwent the following eight treatments; control, 0.2% BAC, 0.75-mg/mL KR-67607, 0.75-mg/mL KR-67607 + 0.2% BAC, 1.5-mg/mL KR-67607, 1.5-mg/mL KR-67607 + 0.2% BAC, 1-mg/mL CBX, or 1-mg/mL CBX + 0.2% BAC. Compounds were topically administered twice daily at 09:30 and 17:30. (**A**) Rose bengal staining after BAC and drug treatments for 10 days. (**B**) Rose bengal scoring analysis. *** *P* < 0.001 vs. control group, ^#^
*P* < 0.05; ^###^
*P* < 0.001 vs. BAC-treated group (*n* = 3/group).

**Figure 3 ijms-21-03729-f003:**
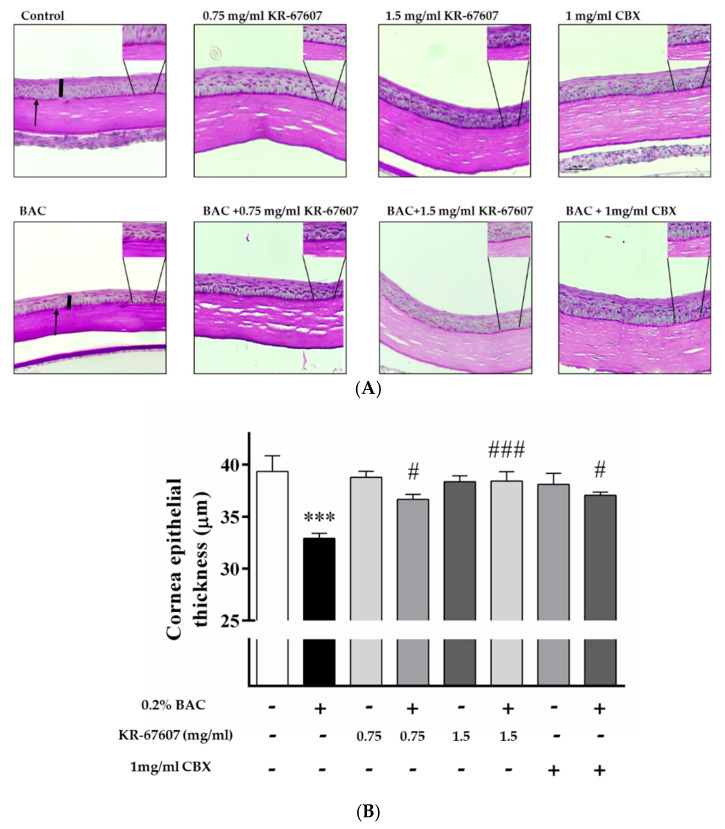
Effects of 11β-HSD1 inhibitors on corneal epithelial and basement membrane thickness in BAC-treated SD rat eyes. Compounds were topically administered twice daily at 09:30 and 17:30. (**A**) Corneal epithelial and basement membrane thickness analysis was measured by PAS staining. Black pillars indicate corneal epithelial thickness; black arrows indicate basement membrane. Images were acquired under 400 × magnification. (**B**) Corneal epithelial thickness measurement. Three portions were (ocular surface nasal bulbar conjunctiva, cornea, and temporal bulbar conjunctiva). Corneal epithelial thickness measured using a DS-Ri2 program. *** *P* < 0.001 vs. control group; ^#^
*P* < 0.05, ^###^
*P* < 0.001 vs. BAC-treated group.

**Figure 4 ijms-21-03729-f004:**
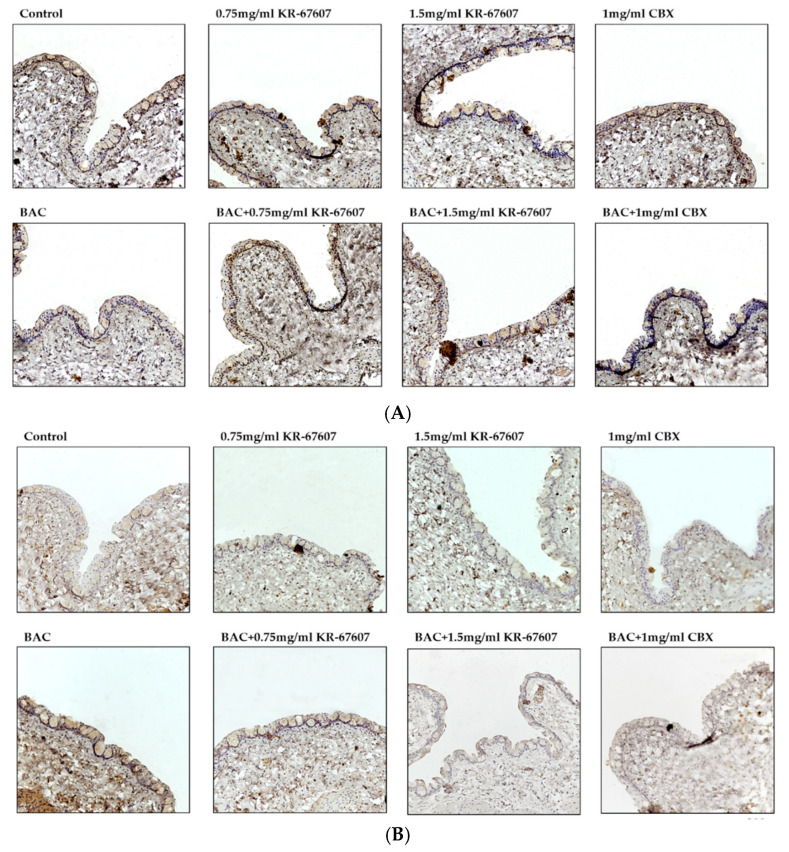
Effects of 11β-HSD1 inhibitors on conjunctival ROS level in BAC-treated SD rat eyes. Compounds were topically administered twice daily at 09:30 and 17:30. SOD1 and 4-HNE expressions were measured by immunohistochemistry staining in control, BAC, BAC+ KR-67607, only KR-67607, BAC + CBX or only CBX-treated conjunctival epithelium. (**A**) SOD1 anti-oxidant enzyme expression in BAC and drug-treated conjunctival epithelium (**B**) Expression of 4-HNE, an ROS byproduct, in BAC and drug-treated conjunctival epithelium. Images were acquired under 200 × magnification.

**Figure 5 ijms-21-03729-f005:**
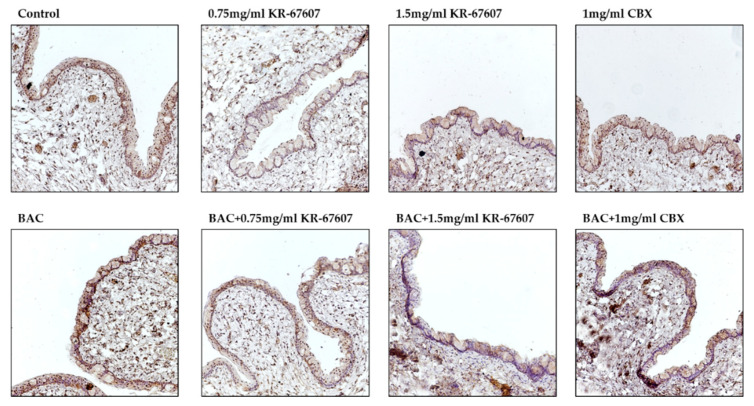
Effects of 11β-HSD1 inhibitors on conjunctival pro-inflammatory marker expression in BAC-treated SD rat eyes. Animals underwent the following eight treatments; control, 0.2% BAC, 0.75-mg/mL KR-67607, 0.75-mg/mL KR-67607 + 0.2% BAC, 1.5-mg/mL KR-67607, 1.5-mg/mL KR-67607 + 0.2% BAC, 1-mg/mL CBX, or 1-mg/mL CBX + 0.2% BAC. Compounds were topically administered twice daily at 09:30 and 17:30. TNF-α expressions were measured by immunohistochemistry staining in control, BAC, BAC+ KR-67607, only KR-67607, BAC + CBX or only CBX-treated conjunctival epithelium. Images were acquired under 200 × magnification.

**Figure 6 ijms-21-03729-f006:**
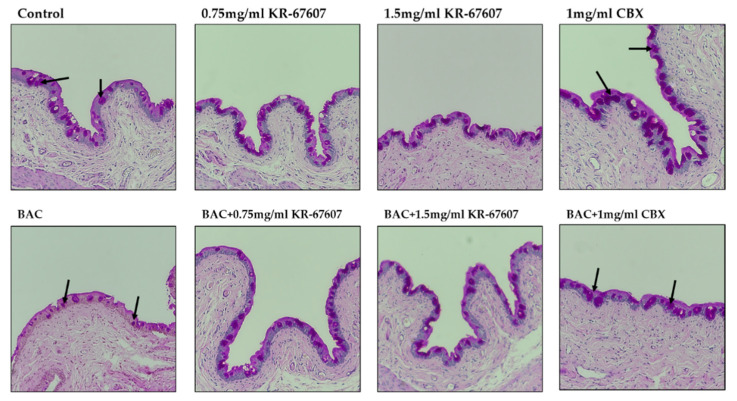
Effects of 11β-HSD1 inhibitors on mucus secretion in BAC-treated SD rat eyes. Animals underwent the following eight treatments; control, 0.2% BAC, 0.75-mg/mL KR-67607, 0.75-mg/mL KR-67607 + 0.2% BAC, 1.5-mg/mL KR-67607, 1.5-mg/mL KR-67607 + 0.2% BAC, 1-mg/mL CBX, or 1-mg/mL CBX + 0.2% BAC. Compounds were topically administered twice daily at 09:30 and 17:30. Analysis of mucus secretion in the conjunctiva was confirmed by PAS staining. The black arrow is pointed at purple circles, which represent the secreted mucus. Images were acquired under 200 × magnification.

**Figure 7 ijms-21-03729-f007:**
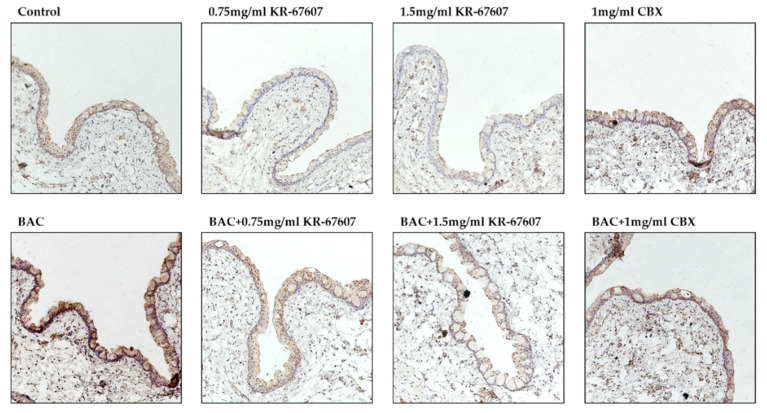
Effects of 11β-HSD1 inhibitors on conjunctival 11β-HSD1 expression in BAC-treated SD rat eyes. Animals underwent the following eight treatments; control, 0.2% BAC, 0.75-mg/mL KR-67607, 0.75-mg/mL KR-67607 + 0.2% BAC, 1.5-mg/mL KR-67607, 1.5-mg/mL KR-67607 + 0.2% BAC, 1-mg/mL CBX, or 1-mg/mL CBX + 0.2% BAC. Compounds were topically administered twice daily at 09:30 and 17:30. 11β-HSD1 expressions were measured by immunohistochemistry staining in control, BAC, BAC + KR-67607, only KR-67607, BAC + CBX or only CBX-treated conjunctival epithelium. Images were acquired under 200 × magnification.

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
