# Peer review of "A Novel Selective 11β-HSD1 Inhibitor, (E)-4-(2-(6-(2,6-Dichloro-4-(Trifluoromethyl)Phenyl)-4-Methyl-1,1-Dioxido-1,2,6-Thiadiazinan-2-yl)Acetamido)Adamantan-1-Carboxamide (KR-67607), Prevents BAC-Induced Dry Eye Syndrome"

_ijms, 2020, doi:10.3390/ijms21103729_

Round 1

Reviewer 1 Report

Dear Authors,

Thank you for presenting a well written manuscript. However, I have commented on a few missing information instances and suggested a few things to improve the reading and reader experience. Please see below. 

  • Section 4.1 explains the concentration of the treatment groups but not the Volume of application?
  • BAC + 1-mg/mL CBX - why is there a discoloration
  • Rose Bengal scoring – can the authors explain how the scoring was done in each of the area? Section 4.2 explains 3 sections each with a maximum of 3 points for damage in the entire area. A breakdown of where the damage was reported would be much more informative.
  • Figure 2A – There seems to be an upper layer present in the control as well as the BAC treated samples, this is not clearly evident (is much lighter in colour) in most of the treatment groups. Can the authors comment on this?
  • In figure 2A BAC image, please consider adding an insert at a higher digital zoom to substantiate the claim made in lines 101-102 “In the group treated with BAC, the basement membrane shape was not formed well, and the dyeing looks pale”. Similar insert in other images would be good too.
  • Have the authors considered including a test for eye performance on the animal while on treatments including the BAC. All histological images show minor and sometimes major differences in the treatment group and control sections. An eye test to demonstrate efficient functioning would be an ideal control. This will make sure the treatments do not cause any damage to eyesight. If this has been done in previous studies, it should be mentioned in the discussion.
  • Results illustrated in figure 5 are not cited in the written text

Author Response

Reviewer 1's comments:

We would like to heartily thank the Reviewer 1 who has kindly examined our manuscript in detail for the purpose of making it better. As we agree with your kind comments, the specified points have been fully dealt with as follows:

1. Section 4.1 explains the concentration of the treatment groups but not the Volume of application?

Answer >> Thank you for your suggestion. Drugs were administrated topically (one drop, 20 ul) twice a day (09:30, 17:30) for 10 days. According to your opinion, we have reflected this dosing volume to 4.1 Animals & drugs administration section, lines 245-246.

2. BAC + 1-mg/mL CBX - why is there a discoloration.

Answer >> After sacrifice, the color of the eye becomes pale, so it is easier to see the ocular surface wound more accurately. It seems that the color is different because there was a slight difference in the time to take the picture. Therefore, the color of the BAC + 1-mg/mL CBX group figure is different.

3. Rose Bengal scoring – can the authors explain how the scoring was done in each of the area? Section 4.2 explains 3 sections each with a maximum of 3 points for damage in the entire area. A breakdown of where the damage was reported would be much more informative.

Answer >> We really appreciate your suggestion. According to Takamura et al (2012, Br J Ophthalmol 2012, 96, (10), 1310-5, reference #51), eye is divided into three part (1, 2 and 3 area). Each compartments were graded as 0 (without any damage), 1 (partial damage), 2 (damage in more than half the area) and 3 (damage in the entire area). This content is written in line 251-255.

< Compartments of eye >

4. Figure 2A – There seems to be an upper layer present in the control as well as the BAC treated samples, this is not clearly evident (is much lighter in colour) in most of the treatment groups. Can the authors comment on this?

Answer >> We agree with your comment. It seems that the conditions for taking the figure are slightly different, so it looks lighter. Thus, we replaced it with a similar figures.

5. In figure 2A BAC image, please consider adding an insert at a higher digital zoom to substantiate the claim made in lines 101-102 “In the group treated with BAC, the basement membrane shape was not formed well, and the dyeing looks pale”. Similar insert in other images would be good too.

Answer >> Thank you for providing these insights. According to your opinion, we added magnified figures of the basement membrane to Figure 2A.

6. Have the authors considered including a test for eye performance on the animal while on treatments including the BAC. All histological images show minor and sometimes major differences in the treatment group and control sections. An eye test to demonstrate efficient functioning would be an ideal control. This will make sure the treatments do not cause any damage to eyesight. If this has been done in previous studies, it should be mentioned in the discussion.

Answer >> We agree with your comment. However, it is difficult to confirm the function of the eyes in our laboratory. We do not know directly whether this drug KR-67607 and CBX affects eyesight and functioning, but, previous our study (Biochem Pharmacol 2019, 169, 113632, reference #35) have confirmed that KR-67607 is non-toxic and ameliorated inflammation and free radicals in human trabecular meshwork cells. In addition, in the glaucomatous animal model, it showed the effect of lowering the cortisol concentration and reducing intraocular pressure in the eyes. These contents were discussed in 3. Discussion section, lines 151-158.

7. Results illustrated in figure 5 are not cited in the written text.

Answer >> Sorry for missing citing it. We have revised the manuscript of 2.5. Result section, line 136.

Reviewer 2 Report

In the present work, the authors tested and claimed that a novel and selective 11β-HSD1 inhibitor called KR-67607 is able to prevent BAC-induced dry eye syndrome. The title matches the contents of the article. The rose bengal staining method was used to confirm the damage to rat ocular surface and in the KR-67607-treated group, the wound healing was marked.

Author Response

1. In the present work, the authors tested and claimed that a novel and selective 11β-HSD1 inhibitor called KR-67607 is able to prevent BAC-induced dry eye syndrome. The title matches the contents of the article. The rose bengal staining method was used to confirm the damage to rat ocular surface and in the KR-67607-treated group, the wound healing was marked.

Answer >> As you mentioned, rose bengal staining can confirm the rat ocular surface damage, and this method is simple and easy to screen. We hope that a novel 11β-HSD1 inhibitor could be used as an ophthalmic treatment. Thank you again your opinion.

Round 2

Reviewer 1 Report

Thank You for taking the time and making changes as suggested.